# Production of the HBc Protein from Different HBV Genotypes in *E. coli*. Use of Reassociated HBc VLPs for Packaging of ss- and dsRNA

**DOI:** 10.3390/microorganisms9020283

**Published:** 2021-01-30

**Authors:** Ivars Petrovskis, Ilva Lieknina, Andris Dislers, Juris Jansons, Janis Bogans, Inara Akopjana, Jelena Zakova, Irina Sominskaya

**Affiliations:** Latvian Biomedical Research and Study Centre, Ratsupites 1, LV-1067 Riga, Latvia; ivars@biomed.lu.lv (I.P.); Ilva.Lieknina@biomed.lu.lv (I.L.); dishlers@biomed.lu.lv (A.D.); jansons@biomed.lu.lv (J.J.); janis@biomed.lu.lv (J.B.); inara@biomed.lu.lv (I.A.); Jelena@biomed.lu.lv (J.Z.)

**Keywords:** HBV genotypes, HBcAg expression, encapsidation

## Abstract

The core proteins (HBc) of the hepatitis B virus (HBV) genotypes A, B, C, D, E, F, and G were cloned and expressed in *Escherichia coli (E. coli*), and HBc-formed virus-like particles (VLPs) were purified with ammonium sulfate precipitation, gel filtration, and ion exchange chromatography (IEX). The best VLP yield was found for the HBc of the HBV genotypes D and G. For the HBc of the HBV genotypes D, F, and G, the possibility of dissociation and reassociation maintaining the native HBc structure was demonstrated. Single-stranded (ss) and double-stranded (ds) ribonucleic acid (RNA) was successfully packed into HBc VLPs for the HBV genotypes D and G.

## 1. Introduction

Ten different hepatitis B virus (HBV) genotypes (A–J) are variably distributed geographically [1,2,3], with the genotypes A and D being the most widespread. The genotypes A, B, C, D, F, H, and I were classified further into subgenotypes with between 4 and 8% intergroup nucleotide differences across the complete genome. HBV genotypes are not only interesting for anthropology and epidemiology but are also useful for clinical reasons. The genotypes C, D, and F are, on average, more pathogenic than the other genotypes, and genotypes A and B respond better to interferon therapy than genotypes C and D. The genotypes can also predict liver disease progression [2,4].

The first reports on the cloning of the full HBV genome date back to the late nineteen-seventies to early nineteen-eighties [5,6,7,8]. With the cloning of the HBV genome and the identification of the HBV S gene (HBs) in 1979, a new era of HBs-based vaccine design and production was entered. In 1986, yeast-derived HBV surface antigen (HBsAg) became the standard recombinant vaccine against HBV [4]. Since the initial expression of gene C (HBV core antigen; HBc) in *Escherichia coli* (*E. coli*) [7], recombinant HBc capsid or virus-like particles (VLPs) have gained acceptance as powerful and prospective scaffolds for recombinant vaccine design (for an exhaustive review, see [9]). This perspective is based on several structural, immunological, and also technological properties of HBc, such as the productivity in heterologous expression systems and immunogenicity of HBc VLPs. HBc was effectively expressed in *E. coli* [10,11] and less effectively in yeasts [12,13,14,15], plants [16,17], and insect cells [18,19].

The recombinant HBc protein possesses the ability to form dimeric units [20,21], which are further self-assembled into two forms of icosahedral particles with T = 4 and T = 3 symmetry [22,23]. For most HBV genotypes, the HBc protein is 183 amino acid (aa) long; however, there are some exceptions—the HBc protein from genotype A is 185 aa long with a DR insertion between aa 153 and 154, but HBc/G is an extraordinary example that differs by its N-terminal DRTTLPYGLFGL extension upstream of the start codon, and thus, it is 195 aa long. The HBc protein is not a large protein, and the VLPs tolerate significant structural modifications while retaining the VLP-forming ability.

Numerous surface modifications of HBc VLPs have been performed, mainly as genetic insertions exploiting the MIR (major immunodominant region) of HBc localized on the types of spikes formed by dimers [24,25,26]. In some cases, the N- and C-termini of HBc have been used for genetic modification. Thus, the N-terminal insertions were expected to be exposed on the VLP surface with a high insertion capacity of up to 120 aa. In this way, vaccine candidates against the foot-and-mouth disease virus (FMDV) [27] and influenza [28,29] were generated.

In contrast to the MIR and N-terminus, the enormous capacity of C-terminal insertions was demonstrated (with up to 559 and even 741 aa sequences from HCV inserted in the C-end [30]), and possible outer exposure of the inserted sequences was shown [31]. The overall praxis demonstrated, however, the internal localization of the inserted aa stretches (see the reviews [9,32]). An original HBc vector in which the C-terminal Arg-residues were fully or partially replaced by Gly residues, allowing the full surface exposure of C-terminal insertions in the HBc C-terminus, was designed [33,34].

A set of HBc vectors was developed for the chemical coupling of foreign epitopes at the tip of the HBc spike residues [35,36,37], avoiding conformational MIR-insertion constraints. Such vectors have been used extensively for the packaging of different oligo- and polynucleotides [38,39]. The packaging of short oligodeoxynucleotides (other than CpG [40]), pregenomic (pg) RNA, and single-stranded (ss) RNA has been also described [41,42]. To a lesser extent, the packaging of ssDNA and, minimally, of double-stranded (ds) DNA has been described as well [41,43].

Packaging into HBc VLPs is possible by different methods, including by non-dissociating methods, such as osmotic shock [43], by VLP treatment with ribonuclease [44], by treatment with micrococcal nuclease [40], by the alkaline treatment of VLPs at high pH [38], and by forced particle dissociation to dimers using guanidine chloride (GuHCl) [41]. The disadvantage of the non-dissociating methods is the contamination of the VLP samples with RNase, and thus, these methods are only usable for the packaging of DNA. The forced in vitro dissociation and free reassociation of HBc VLPs from dimers allows removing material incorporated within VLPs—mostly, RNA of host origin—producing so-called “empty” HBc VLPs. Reassociation in the presence of an appropriate target substance allows incorporation within HBc VLPs as well.

The production and purification of unmodified recombinant HBc VLPs as the first step, with their further chemical modification, is generally a more attractive approach in vaccine design, since the integrity and structural stability of genetically modified HBc VLPs can differ considerably depending on the physical/chemical properties of the particular foreign insert. Unmodified HBc VLPs produced in *E. coli* can be more easily purified by fast non-sophisticated protocols, and their use for substrate packaging is more promising, advocating for a deeper technological investigation of different HBc proteins to find the most promising candidates for further applications in vaccine design and the development of genetic delivery tools.

In this study, we used HBc proteins from different HBV genotypes for their expression in *E. coli* and showed that HBc from genotypes D and G were the most effectively produced and that their corresponding VLPs were easily purified; therefore, we propose that these two types of HBc are the most suitable for biotechnological in vitro applications, such as the development of HBc-based delivery tools.

## 2. Materials and Methods

### 2.1. Bacterial Strains

Two *E. coli* strains—*E. coli* K802 (F^-^, lacY1, or (cod-lacI)6, glnX44(AS), galK2(Oc), galT22, e14 mutant, mcrA0, rfbC1, metB1, mcrB1, hsdR2) and BL21 (F^–^, ompT, hsdSB (rB^–^, mB^–^), dcm, gal) were used to produce HBc VLPs.

### 2.2. Construction of Plasmids for the Expression of HBc

Regions encoding aa 1–185 of the HBc from the HBV genotype A (GenBank accession number MW401519) and aa 1–183 of the HBV core from genotypes E (GenBank accession number MW401520) and F (GenBank accession number KY458062) were amplified using the oligonucleotides 5′-GGCCATGGACATTGACCCTTA-3′ and 5′-GTAGAGGATCCTTACTAACATTGGGAA-3′ as the forward and reverse primers, respectively, and were inserted into the NcoI/BamHI pHBc183NcoI (HBc/D) plasmid after treatment with the endonucleases NcoI and BamHI [45].

The region encoding aa 1–195 of HBc from genotype G (GenBank accession number JQ707677) was amplified using the oligonucleotide 5′-GTGGAATTGGCCATGGATAGA-3′ as the forward primer and the primer used to amplify the HBc of genotypes A, E, and F as the reverse primer. The regions encoding the HBc of HBV of genotype B (GenBank accession number MW401525) and genotype C (GenBank accession number MW401526) were obtained by gene synthesis (SGI-DNA, USA), and both contained the flanking sites for NcoI and BamHI, and were inserted into a NcoI/BamHI pHBc183NcoI (HBc/D) plasmid after treatment with the endonucleases NcoI and BamHI.

### 2.3. Screening of Transformants, Expression of HBc, and Purification of HBc VLPs

Two media were used in this study for the expression of the HBc gene under a Ptrp promoter: *E. coli* producer cells were cultivated in (1) Trp-deficient enriched minimal M9-Cas medium, which is the standard M9 salt medium supplemented with acid-hydrolyzed Casamino Acids (10 g/L, BD, USA), glucose (2 g/L), and Km (10 mg/L), or (2) in phosphate-buffered 2×TYP medium supplemented with phosphates (3.47 g of KH_2_PO_4_, and 18.8 g of K_2_HPO_4_ per liter, pH 7.4), glucose (2 g/L), and Km (10 mg/L). The screening of the transformants was performed as follows: 10 individual transformant colonies from selective LB-agar plates (LB agar supplemented with Ap at 50 mg/L) were picked and transferred into ten tubes (9 *×* 150 mm, equipped with cotton plugs) containing 5 mL of LB medium (with Ap added to 20 mg/L) and incubated overnight (~16–18 h) at 37 °C without agitation.

These cultures served as primary seed cultures for further experiments following the screening experiments using both M9-Cas and 2×TYP media. For this, the seed culture was diluted 1:100 in 5 mL, and the tubes were incubated on a shaker at a 45° angle at 37 °C overnight. The final OD_540_ in the M9-Cas medium reached 3–5, and that in 2×TYP, 8 to 10 OD_540_ units. For the analysis of HBc expression, normalized amounts of cells (two OD_540_ units) were taken for sodium dodecyl sulphate–polyacrylamide gel electrophoresis (SDS-PAGE), and the gels were scanned using the software GelAnalyzer2010. The best clone was taken for the next stage—semi-preparative cultivation using 750 mL half-round flasks filled with 300 mL of medium. The expression cultures were incubated on an orbital shaker at 200 rpm and 37 °C for 14–16 h. The final OD_540_ reached 5–8 in the M9-Cas medium and 8–10 in the 2×TYP medium.

The purification of HBc VLPs was generally performed with a combination of gel filtration on a Sepharose4FF column and ion exchange chromatography on a Fractogel DEAE (M) column. Eight grams of frozen (−20 °C) biomass was thawed on ice, resuspended in four volumes of lysis buffer (50 mM TrisHCl, pH 8.0. 5 mM EDTA, 0.5 mM PMSF, 150 mM NaCl, 0.1% Triton X100, and 5 mM DTT), and disintegrated using a French Press (FrenchPress, GlenMills Inc., Clifton, NJ, USA) at 20,000 psi for three cycles. The total yield of the target protein was verified using SDS-PAGE. For a better yield of the VLP, urea was added up to 0.5 M to the cell lysate, followed by incubation with DNaseI (125 µg/mL) and MgCl_2_ at 10 mM at +4 °C for 30 min. Clarification was carried out at 10,000 rpm (13,000× *g*), +4 °C for 30 min. To the supernatant, ammonium sulfate was added to 10% saturation, followed by incubation on a rotator at +4 °C for 1 h. After clarification at 10,000 rpm (13,000× *g*), +4 °C for 30 min, ammonium sulfate was added to 35% saturation to the supernatant, followed by incubation on a rotator at +4 °C for 1 h. After centrifugation at 10,000 rpm (13,000× *g*), +4 °C for 30 min, the pellet was dissolved in PBS with 0.5 M urea, 0.5 mM PMSF, 0.1% Triton X100, and 5 mM DTT. Before loading on a Sepharose4FF 320 mL column, the solution was clarified at 10,000 rpm (13,000× *g*), +4 °C for 30 min. The fractions containing VLPs were loaded on an Fractogel DEAE (M) 60 mL column equilibrated with PBS/5 mM DTT. The VLPs were eluted from the column in a gradient of 1M KCl, and the fractions were analyzed using SDS-PAGE and native agarose gel electrophoresis (NAGE).

### 2.4. Electron Microscopy and Dynamic Light Scattering Analysis of HBc VLPs

VLP-containing fractions from the column or VLP preparations in PBS were subjected to electron microscopy (EM) analysis immediately or after storage at +4 °C for 1–2 days. Samples were adsorbed to carbon-formvar-coated copper grids and negatively stained with a 1% aqueous solution of uranyl acetate. The grids were examined at 100 kV using a JEM-1230 electron microscope or a JEM-100C electron microscope at 80 kV (both from Jeol Ltd., Tokyo, Japan). Dynamic light scattering (DLS) was used to determine the size distribution profiles of the particles and performed on a Zetasizer Nano ZS instrument (Malvern Instruments Ltd., Malvern, UK). The DLS results were analyzed using the DTS software (Malvern, version 6.32). The ImageJ program was used for precise measurements of the HBc particle sizes.

### 2.5. ssRNA and dsRNA Used for Packaging

ssRNA of P. aeruginosa RNA-bacteriophage PP7 was extracted from purified phage PP7 (ATCC 15692-B2) particles. For the extraction of the RNA, 15 mL of TRI Reagent^TM^(Sigma-Aldrich, T9424, St. Louis, Missouri, USA), 150 µL of 10% SDS, and 3 mL of chloroform were added to the phage solution with 3 mg of phage particles in 3 mL of PBS. Further steps were performed according to the manufacturer’s protocol. The final concentration of isolated ssRNA was 0.81 mg/mL.

The dsRNA used in this work is the functional ingredient of Larifan [46]; this was a kind gift from Dr. Med. Guna Feldmane (Larifāns Ltd., Riga, Latvia). The mass of this kind of dsRNA is approximately 500 kDa, and the average length is 400 nt. The source of the dsRNA was non-suppressor *E. coli* Su^-^ cells infected with the RNA-phage f2 coat protein amber mutant f2sus11 [47,48]. The concentration of dsRNA used in this study was 10.0 mg/mL.

### 2.6. Dissociation/Reassociation of HBc VLPs

For the dissociation of HBc VLPs, GuHCl- and LiCl- containing buffer was used [41]. After incubation at +4 °C for 16 h and centrifugation at 10,000 rpm (13,000× *g*) at +4 °C for 30 min to remove debris, the supernatant was applied on a Superose6/120 mL column, equilibrated with dissociation buffer. Fractions containing dimers were dialyzed against PBS using a 14 kD membrane (Sigma-Aldrich, Cat. No: D9777-100FT, St. Louis, MO, USA) at +4 °C for 12 h. The protein solution was centrifuged at 10,000 rpm (13,000× *g*) for 30 min. The yield of dimers was calculated as a proportion of the HBc material in the dimer fractions after the analysis of the dissociation products on Superose6 and the amount of HBc VLPs used for dissociation.

Reassociation of the HBc dimers occurred during dialysis. The dissociation buffer was changed to PBS twice with an 8 h interval. The obtained reassociated capsids were tested using SDS-PAGE, NAGE, electron microscopy, and DLS.

### 2.7. Packaging of ssRNA and dsRNA in HBc/D VLPs

The packaging of ssRNA and dsRNA in HBc/D VLPs was achieved by the reassociation of HBc/D dimers in the presence of appropriate RNA as a packaging substrate. For the packaging of PP7 ssRNA, 5 mL of HBc/D dimers (1 mg/mL) were mixed with 1.5 mL of ssRNA (0.81 mg/mL). After reassociation, 50 µL of Benzonase (25 U/µL, EMD Chemicals (San Diego, CA, USA). was added, followed by incubation at RT overnight. The reassociated VLPs were purified by gel filtration on a Superdex200 120 mL column equilibrated with PBS. The VLPs were stored at +4 °C.

For the packaging of dsRNA in HBcAg/D VLPs, 15 mL of HBcAg/D dimers (1 mg/mL) and 1 mL of dsRNA (10 mg/mL) were used in a reassociation reaction. After dialysis, the protein was concentrated into 3 mL, with a final concentration of VLPs of 5.0 mg/mL. Then, 100 µL of Benzonase (25 U/µL) was added, followed by incubation at RT overnight. Reassociated HBc/D VLPs with packaged dsRNA were purified from contaminants on a Superdex200 120 mL column equilibrated with PBS. The VLPs were stored at +4 °C.

### 2.8. Packaging of dsRNA in HBcAg/G VLPs

The packaging of dsRNA in HBC/G VLPs was achieved by the incubation of the reassociated capsids in a low-ionic-strength medium. Four and a half milligrams of reassociated HBc/G VLP material (1.5 mg/mL) was added to 4.5 mg of dsRNA (0.167 mg/mL, in water). The total volume was 30 mL. After incubation at RT for 16 h, the mixture was concentrated into 3 mL, and 50 µL of Benzonase (25 U/µL) was added. The mixture was incubated at RT for 16 h. VLPs with packaged nucleic acids were purified on a Superdex200 120 mL column equilibrated with PBS. The VLPs were stored at +4 °C.

### 2.9. Extraction of RNA from HBc VLPs

For the extraction of RNA from HBc particles (VLPs), the disruption of the particles with Proteinase K was used. Briefly, to 160 µg of HBc/D VLP protein solution native or packaged with dsRNA, SDS up to 0.5% and 10 µL of Proteinase K (Thermo Scientific EO0491, Vilnius, Lithuania) were added, and the mixtures were incubated at 37 °C overnight. The samples were deproteinized with chloroform, and the RNA was dissolved in 20 µL of dH_2_O. Ten microliters of the solution was used for analysis by NAGE.

## 3. Results

### 3.1. Construction of Expression Plasmids and Purification of HBc VLPs

In this study, the expression of the HBc genes of HBV of genotypes A, B, C, D, E, F, and G was investigated (Figure 1).

Expression vectors were constructed on the basis of the pBR327 vector and *E. coli* Ptrp promoter (Figure 2).

Transformed *E. coli* cells were cultivated under the self-induction of the Ptrp promoter without the use of an inducer. The expression of an appropriate HBc gene was compared in two *E. coli* strains—in K802 (K-12 lineage) and BL21 (a derivative of *E. coli* B). For the cultivation of producer cells, two alternative media were generally used—the Trp-rich 2xTYP (phosphate buffered 2TY) and Trp-deficient M9-Cas (M9 medium supplemented with acid-hydrolyzed casein; see Materials and Methods, Section 2.3). In the course of these comparisons, the optimal variant was selected, ensuring the best specific expression level of the target protein (the level of the HBc protein per gram of cells) and the best solubility of the expressed HBc protein (the level of HBc in the soluble fraction/total HBc synthesized).

The optimal host strain and medium combination was determined individually for each particular HBc protein originating from different HBV genotypes. The best variant (combination of host strain and medium) found by analytical cultivation (using 5 mL cultures in tubes) was further used for scale-up cultivation to obtain several grams of biomass. The expression levels in individual transformants were also compared (Table 1), and the best transformants were used for further cultivation, first, for analytical cultivation in both media, and then, for preparative cultivation (to obtain several grams of biomass) using the optimal medium.

The production of HBc VLPs for the genotypes A, B, F, and G was performed in *E. coli* BL21 cells, and that for the HBc of the genotypes C, D, and E, in *E. coli* K802. In most cases, 2×TYP medium was used, except for the HBc of genotypes E and F, where the M9-Cas medium was found to be optimal. The level of HBc production varied between different genotypes and individual transformants, as detected via the zone intensities of the scanned SDS-PAGE gels (Table 1). Although the expression level between the transformants varied in all the HBc genotypes with a 1.5–2 times range, the general expression levels for all the genotypes were evaluated as being approximately the same. However, the final yields of the HBc VLPs from different genotypes differed remarkably after purification.

VLP formation was observed in all cases, and the HBc protein was purified in the form of VLPs from the fraction of soluble proteins after the extraction procedure. The purification procedure included sedimentation with ammonium sulfate and two-step chromatography—gel filtration followed by anion exchange. The purity of the VLPs was tested using SDS-PAGE. The presence of nucleic acids was tested using NAGE and staining with EtBr (Figure 3A).

The protein concentration of the purified HBc in the form of VLPs was detected by the Bradford protein assay, and the total yield of the HBc VLPs was calculated (Table 1). The highest yield of HBc VLPs (as mg/g of wet cells) was observed for the HBc of genotypes D and G. All the HBc VLPs contained packaged nucleic acids (Figure 3A, line 6). The characterization of the purification process is given in Figure 3, and a summary of the HBc expression/purification data is presented in Table 1.

A higher purity of HBc correlated with a higher yield of VLPs. Electron microscopy revealed the domination of T4 particles in all the cases of expressed and purified HBc VLPs. The HBc particle size determined using the ImageJ program was, on average, 34.4 nm for ten T4 particles and 27.75 nm for ten T3 particles. The hydrodynamic diameters of the HBc particles detected using DLS were found to be around 38 nm.

### 3.2. Dissociation and Refolding of HBc VLPs of Genotypes D, G, and F

HBc/D, HBc/G, and HBc/F VLPs were selected for the dissociation–reassociation of HBc VLPs. In this study, VLPs were dissociated using dissociation buffer containing GuHCl and LiCl. After incubation in the GuHCl-containing buffer, part of the VLPsremained in sediment. After the removal of the debris, the supernatant was loaded on a Superdex200 column to separate the dimers from the non-dissociated capsids and other impurities. The results of the representative separation are shown in Figure 4. The results show that the supernatant mostly contained dimers (Figure 4). The most effective dissociation was observed for HBc/G (80% of the treated soluble VLPs as dimers), followed by HBc/D (70%) and HBc/F (40%). The dimer purity as determined by SDS-PAGE was generally more than 90% for all three types of HBc.

The reassociation of dimers into VLPs occurred under dialysis against PBS, resulting in “empty” HBc particles. The characterization of the reassociated HBc particles is given in Table 2 and Figure 5. The reassociation was highly effective in all cases, with 80–90% of the dimers being reassociated in VLPs after overnight dialysis against PBS. Under storage (+4–6 °C), the dimers were stable for only few days, but the reassociated capsids were stable for more than a year in PBS (with 0.5 M NaCl) for all three types of HBc.

### 3.3. Packaging of ds- and ssRNA into HBc VLPs for Genotypes D and G

The results of the dsRNA and ssRNA packaging are shown in Figure 6 and Figure 7, respectively. In the case of HBc/D, packaging was achieved by the reassociation of the HBc/D dimers in the presence of RNA (see Materials and Methods, Section 2.7). In the case of HBc/G, packaging was achieved by the incubation of reassociated HBc/G capsids in a low-ionic-strength medium in the presence of dsRNA (see Materials and Methods, Section 2.8).

### 3.4. Analysis of RNA Packaged in HBc VLPs

For the analysis of the nucleic acid content of the HBc VLPs, native HBc/D particles and HBc/D particles obtained after the reassociation of dimers in the presence of dsRNA were treated with Proteinase K to disrupt the particles (Materials and Methods, Section 2.9). After deproteinization, the obtained nucleic acids were analyzed by NAGE. As shown in Figure 8, RNA from both the native and packaged particles was extracted in comparable amounts (tracks 2 and 3). Similar results were obtained for the HBc/G packed with dsRNA (not shown). The particles obtained by the reassociation of dimers are free of RNA (track 6, no signal), and these particles are not able to enter the gel, staying at the top (track 7).

## 4. Discussion

In this study, we compared the production levels of the HBc from the HBV genotypes A, B, C, D, E, F, and G in *E. coli* cells and used the best producers to obtain high-quality VLPs suitable for packaging nucleic acids. To demonstrate the packaging potential of HBc-formed icosahedral particles—VLPs—we used two types of RNA from two different RNA-phages: the native RNA extracted from *P. aeruginosa* RNA-phage PP7 particles and dsRNA extracted from non-suppressor *E. coli* Su^-^ cells infected with *E. coli* RNA-phage f2 nonpolar coat protein amber-mutant f2sus11 (for the properties of polar and nonpolar mutants, see [49]).

Here, both types of RNA were packaged in VLPs formed by HBc from the HBV genotypes D and G as the most prospective HBc candidates in technological aspects. For packaging, the HBc VLPs were fully purified by the dissociation of the VLPs to dimers, allowing the removal of incorporated “impurities” of host cell origin (mainly RNA). Two methodologies were used to further package the target substrate within the HBc particles: (a) the reassociation of HBc/D dimers in the presence of phage RNAs—native ssRNA (here referred to as “PP7 ssRNA”) and dsRNA from phage f2—and (b) the incorporation of phage f2 dsRNA within the HBc “empty” particles obtained after the reassociation of HBc/G dimers.

We chose RNA from RNA-phages for two reasons: (i) ssRNA is a source of functional RNA molecules useful for testing the protective effects of the HBc VLPs (we will describe the protective effect of the HBc VLPs in a separate manuscript [50]), and (ii) dsRNA is an interferon inductor and substrate with potentially immunomodulatory properties, a functional ingredient of the pharmaceutical product *Larifan* [46,47,51] (for an updated exhaustive review on *Larifan,* see [48]).

The expression of the appropriate HBc genes was compared in two *E. coli* strains: K802 and BL21. For the cultivation of producer cells, two alternative media—phosphate buffered 2×TYP and M9Cas—were used. The best combination of host strain and medium in each case was found by analytical cultivation. VLP formation was observed in all the cases; however, the production level for HBc itself and the VLP yield for HBc proteins of different origin varied between the best and the worst HBc producers by four to five times. The best yield of VLPs was found for the HBc from the HBV genotypes D and G, where the HBc/G producer demonstrated the best expression stability at the transformant level (Table 1).

Our non-sophisticated purification of HBc VLPs included precipitation with ammonium sulfate and a combination of gel filtration and ion exchange chromatography. We speculate here that, for VLP formation, the prolonged HBc/G protein has certain advantages, and this is likely to be the reason for the high HBc production stability observable at the transformant level (Table 1). All our previous experience with recombinant HBc VLPs was based on HBc/D VLPs [33,45], and here, we found that HBc/D particles could be easily and effectively dissociated to dimers to incorporate nucleic acid material in the process of dimer reassociation, which was demonstrated here with the incorporation of phage RNA in HBc/D VLPs.

Although HBc/D is the next technologically best HBc protein after HBc/G, the expression level varied significantly between the HBc/D-transformants (Table 1), and the regular selection of the best clone for the production of HBc/D VLPs was always necessary. We propose here that HBc/G VLPs have definite technological advantages over HBc/D VLPs due to their stable and high production levels in *E. coli* cells, allowing sufficient amounts of VLPs to be obtained (around 10 mg from 1 g of cells) with simple lab-scale protocols. As we showed here, HBc/G VLPs along with HBc/D VLPs can be impressive carriers of foreign functional RNA molecules.

The weakest HBc producer in our screening tests was found to be the producer of the HBc from genotype C, also demonstrating that a higher production of HBc ensures a higher purity of HBc VLPs as the final product. Thus, although the DLS data for all the HBc preparations were similar, showing good size homogeneity within the VLP population, the quality of the VLP preparations was very different: the EM data revealed a high VLP instability in the HBc/C and HBc/B preparations (Figure 3).

The dissociation/reassociation of HBc VLPs has been demonstrated before [52], and here, we demonstrated the successful dissociation/reassociation of VLPs formed by the HBc from HBV genotypes D, G, and F. Based on the technological advantages of HBc/D and HBc/G VLPs, we propose that both these types of HBc are suitable and effective for obtaining “pure” VLPs to be used for packaging functional substrates, such as the RNAs used here in this work. We suggest that these technological findings will promote further HBc VLP applications in the generation of prophylactic and therapeutic vaccines and gene therapy tools.

## Figures and Tables

**Figure 1 microorganisms-09-00283-f001:**
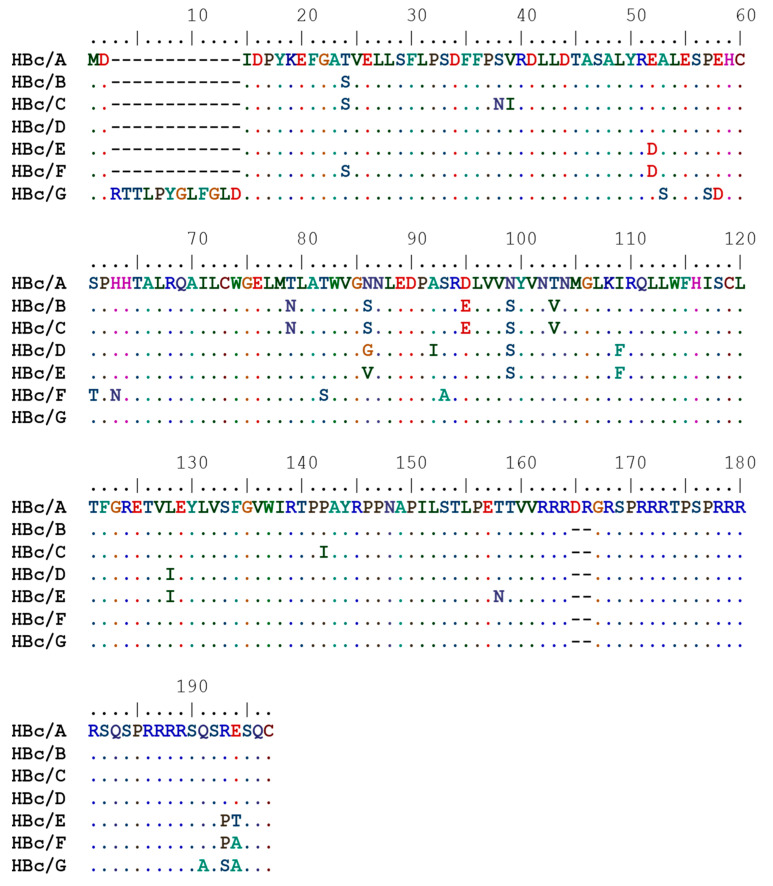
Amino acid sequences of the core proteins (HBc) of hepatitis B virus (HBV) genotypes A, B, C, D, E, F, and G used in this work.

**Figure 2 microorganisms-09-00283-f002:**
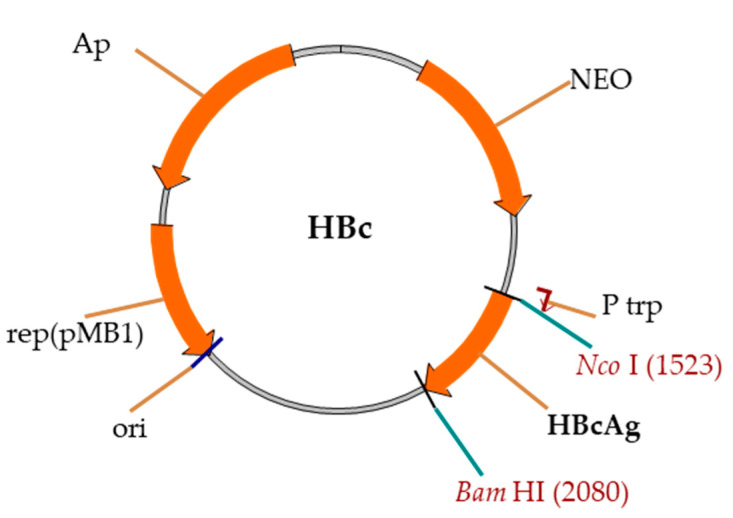
Schema of the pBR327-based vector for the expression of HBc genes. Rep(pMB1)—ori region, Ap—gene for Ap resistance, NEO—gene for neomycin–kanamycin resistance, Ptrp—*E. coli* Ptrp-promoter, and HBc—HBc gene.

**Figure 3 microorganisms-09-00283-f003:**
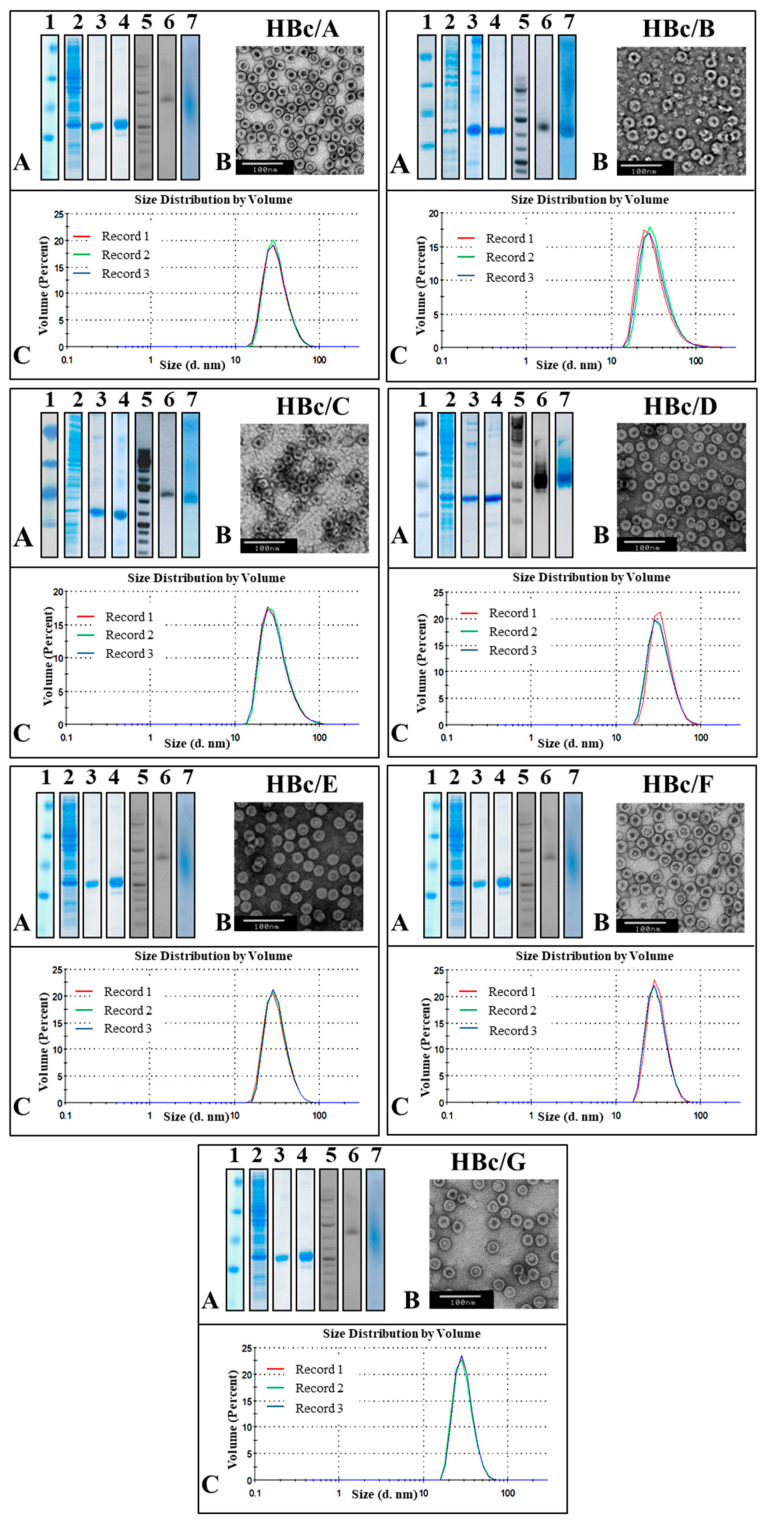
Representation of the purification process for virus-like particles (VLPs) formed by HBc from different HBV genotypes. (**A**) **1**–**4**—SDS-PAGE, stained with Coomassie Brilliant Blue R-250: **1**—Protein Molecular Weight Marker (26612, Thermo Scientific™, Vilnius, Lithuania); **2**—*E. coli* cells before HBc purification; **3**—the peak fraction from *Sepharose* 4FF; **4**—peak fraction from *Fractogel* DEAE (M). Note: HBc/E and HBc/F proteins were purified automatically by two-step chromatography. **5**–**7**—NAGE: **5**—GeneRuller 1 kb DNA Ladder (SM1331, Thermo Scientific™, Vilnius, Lithuania); **6**, **7**—analysis by NAGE of final VLP product: **6**—gel stained with EtBr; **7**—gel stained with Coomassie Brilliant Blue R-250. (**B**) Electron microscopy of purified VLPs. (**C**) DLS analysis of final product.

**Figure 4 microorganisms-09-00283-f004:**
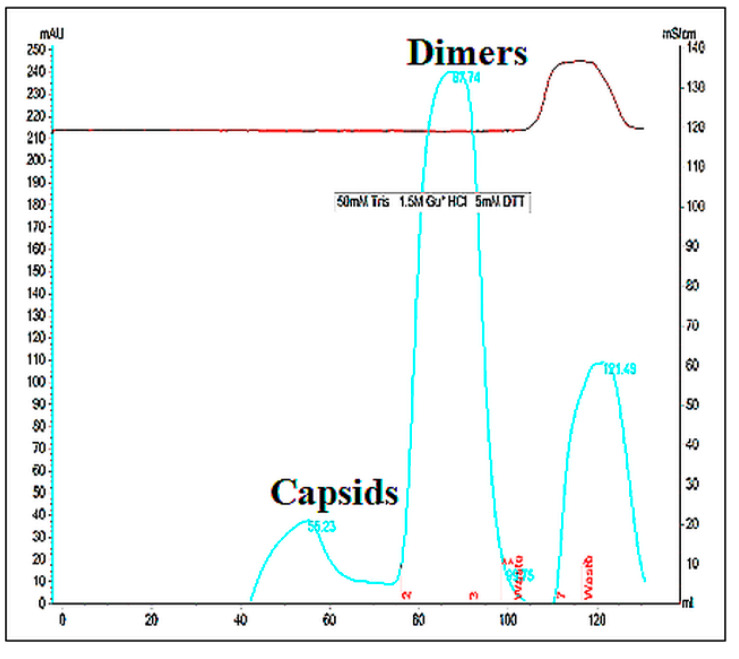
Analysis by Superdex200 gel filtration of HBc/G VLPs after the incubation of VLPs in dissociation buffer containing GuHCl/LiCl and removal of insoluble material.

**Figure 5 microorganisms-09-00283-f005:**
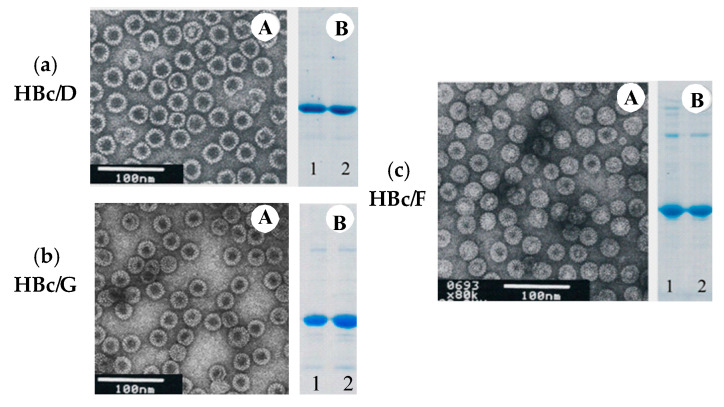
Analysis of the reassociated VLPs: HBc of genotype D (**a**), genotype G (**b**), and genotype F (**c**): **A**—electron microscopy of the reassociated VLPs; **B**—SDS-PAGE of the HBc dimers before reassociation (**1**), and SDS-PAGE of the VLPs after the reassociation (**2**).

**Figure 6 microorganisms-09-00283-f006:**
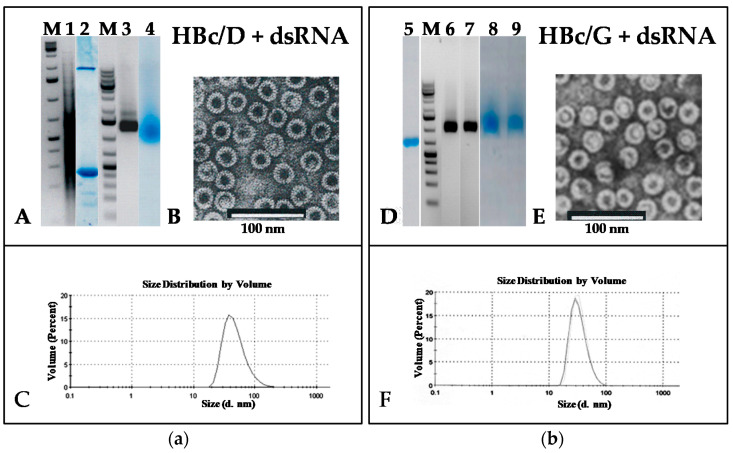
Analysis of HBc/D (**a**) and HBc/G (**b**) VLPs packaged with phage f2sus11 dsRNA. (**a**) **A**, **M**—NAGE in 2% agarose of GeneRuller 1 kb DNA Ladder (SM1331, Thermo Scientific™, Vilnius, Lithuania) and **1**—of f2sus11 dsRNA; **2**—SDS-PAGE of the *Superdex* 200 peak fraction containing VLPs packaged with dsRNA, stained with Coomassie Brilliant Blue R-250; **3**, **4**—NAGE of the same *Superdex* 200 fraction: **3**—stained with EtBr, and **4**—stained with Coomassie Brilliant Blue R-250. **B**, electron microscopy of HBc/D VLPs packaged with dsRNA. **C**, DLS of HBc/D VLPs packaged with dsRNA. (**b**) **D**, **5**—SDS-PAGE of HBc/G VLPs packaged with dsRNA; **M**—NAGE of GeneRuller 1 kb DNA Ladder (SM1331, Thermo Scientific™, Vilnius, Lithuania); **6**–**9**—NAGE of HBc/G VLPs packaged with dsRNA, stained with EtBr (**6**—native HBc/G VLPs; **7**—packaged HBc/G VLPs), and stained with Coomassie Brilliant Blue R-250 (**8**—native HBc/G VLPs; **9**—packaged HBc/G VLPs). **E**, electron microscopy of HBc/G VLPs packaged with dsRNA. **F**, DLS of HBc/G VLPs packaged with dsRNA.

**Figure 7 microorganisms-09-00283-f007:**
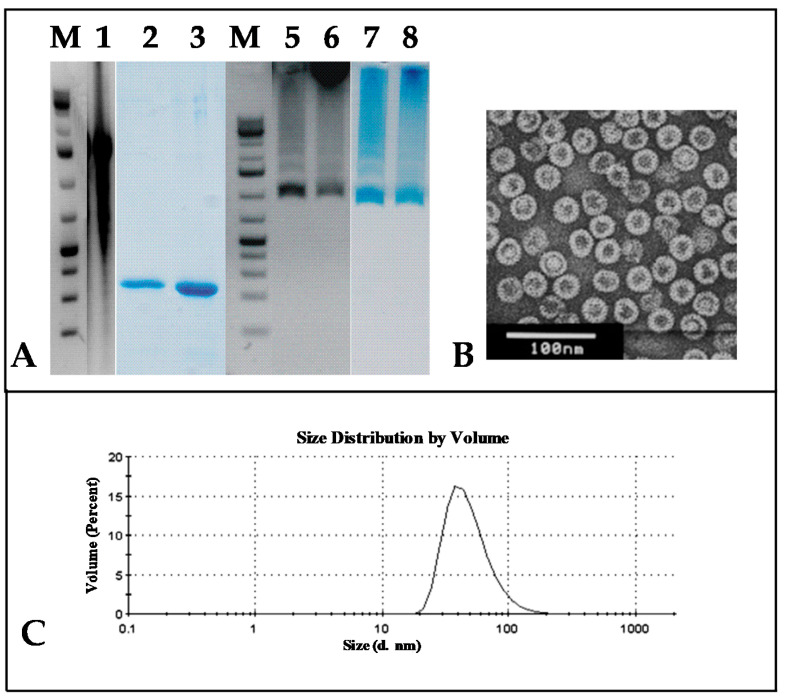
Analysis of HBc/D VLPs packaged with phage PP7 ssRNA. (**A**) **M**—NAGE in 2% agarose of GeneRuller 1 kb DNA Ladder (SM1331, Thermo Scientific™, Vilnius, Lithuania) and **1**—of phage PP7 RNA; **2**, **3**—SDS-PAGE of packaged VLPs stained with Coomassie Brilliant Blue R-250; **5**–**8**—NAGE of packaged HBc/D VLPs stained with EtBr; **3**, **5**, **7**—supernatant of the packaged VLP preparation; **2**, **6**, **8**—debris of the packaged VLP preparation. (**B**) Electron microscopy of the supernatant of the packaged VLP preparation. (**C**) DLS of the supernatant of the packaged HBc/D VLPs. HBc/D VLPs with packaged phage PP7 ssRNA.

**Figure 8 microorganisms-09-00283-f008:**
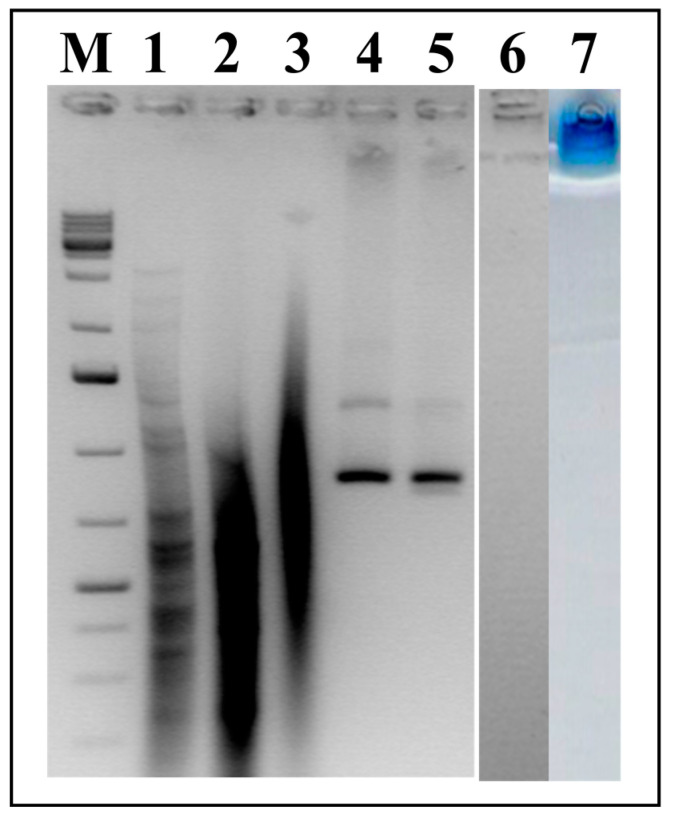
Extraction of RNA from HBc/D VLPs; NAGE in 2% Agarose. **M**—GeneRuller 1 kb DNA Ladder (SM1331, Thermo Scientific™, Vilnius, Lithuania); **1**—dsRNA used for packaging; **2**—RNA extracted from packaged HBc/D VLPs; **3**—RNA extracted from native HBc/D VLPs; **4**—native HBc/D VLPs; **5**—HBc/D VLPs packaged with dsRNA; **6**, **7**—empty HBc/D VLPs; **M** and **1**–**6**—stained with EtBr; **7**—stained with Coomassie Brilliant Blue R250. Note: different amounts of VLPs were loaded into tracks **4** and **6**, not corresponding to the amounts of VLPs used for the extraction of RNA (tracks **2** and **3**).

**Table 1 microorganisms-09-00283-t001:** Characterization of expression of HBc proteins from different HBV genotypes. The HBc purity is shown for the expression culture obtained from the best transformant.

HBc Origin (HBV Genotype)	HBc Length, aa	Variability of HBc Expression Level in Five Individual *E. coli* Transformants (% of Total Protein)	HBc Purity, %	VLP Outcome, mg/g Wet Cells
A	185	3.1–7.6	>90	8.45
B	183	4.1–7.1	80	5.6
C	183	3.3–4.9	70	4.4
D	183	4.2–10.0	≥90	15
E	183	5.4–6.3	80	8.4
F	183	4.5–5.2	80	7.4
G	195	5.6–7.2	>90	20

**Table 2 microorganisms-09-00283-t002:** Summary of the dissociation and reassociation of HBc VLPs from the genotypes D, F, and G.

Parameter	HBc/D	HBc/G	HBc/F
Dimer content	30%	30%	30%
Reassociated dimers	80%	90%	80%
VLP purity after reassociation	97%	97%	97%
VLP stability after reassociation	>15 months	>15 months	>15 months
VLP size, nm (dynamic light scattering, DLS)	38	38	38

## Data Availability

Not applicable.

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
