# Peer review of "Production of the HBc Protein from Different HBV Genotypes in E. coli. Use of Reassociated HBc VLPs for Packaging of ss- and dsRNA"

_microorganisms, 2021, doi:10.3390/microorganisms9020283_

Round 1

Reviewer 1 Report

I read with interest the manuscript of Petrovskis and collaborators entitled “Production of HBc protein from different HBV genotypes in E. coli. Use of reassociated HBc VLPs for packaging of ss and ds RNA”.

The focus of the study was to produce viral like particles suitable for packaging of biological substrates such as nucleic acids. To this end, authors transformed E. Coli with pBR327 plasmid containing the cloned sequence encoding for protein HBc of the HBV core, referable to several HBV genotypes (A, B, C, D, F and G). After evaluation of the HBc expression in terms of quantity and specificity for each genotype, purified HBc characterized from genotypes D, F and G have been subjected to dissociation/reassociation process in presence of ssRNA or dsRNA molecules. Based on data about particle size and structure stability obtained by DLS and EM analyses, authors conclude that VLPs from HBc/D and HBc/G produced by the proposed method can be used for packaging of RNAs and are suitable as delivery tools for biological application related to gene therapy or vaccination.

The manuscript would benefit of a through-out editing of the English writing style. Anyway, the study is methodologically sound. Introduction and discussion are exhaustive. Materials and methods are explained in detail, results and conclusions are well supported by experimental data.

Suggestions for manuscript correction are:

Line 101: “Region encoding aa 1-195 of HBc from genotype G 99 (GeneBank accession number JQ707677) was amplified using oligonucleotide 5'-GTGGAATTGGCCATGGATAGA-3' as forward primer.” It is not reported the reverse primer. Specify if it is the same primer used to amplify HBc for genotypes A, E and F or write the oligonucleotide sequence.

Line 113:  A full stop is missed at the end of the sentence.

Line 118: The comma at the end of the sentence should be corrected with a full stop.

Line 126: Correct the following sentence “The expression cultures were incubated fat 37° for 14–16 h…” as “The expression cultures were incubated at 37°C for 14–16 h…”.

Line 136: Concentration of DNAse I is expressed as [γg/ml]. Considering that 1γg is equal to 1μg could be better to change it to [μg/ml] that is more widely used.

Line 137- 144: The purification of extracted VLPs by several washing steps (incubation and centrifugation) is difficult to follow.  Report the procedure using an easier for readers.

Line 149: The author reports that VLPs in PBS were used for electron microscopic analysis. Specify how long after elution of the VLPs the EM analysis was performed or if the VLPs were fixed with glutaraldehyde to avoid alteration of the structure.

Line 160:  A full stop is missed at the end of the sentence.

Line 163: Delete the round brackets before the comma.

Line 177: Correct the following typo “reasociation” as “reassociation”.

Line 176: Paragraph 2.7 "Reassociation of HBc VLPs" is only one and a half lines long. Perhaps you should add it to paragraph 2.6 and make a single paragraph entitled "Dissociation/reassociation of HBc VLPs".

Line 196: Correct the following typo “dsRNA (0.167 mg/ml, of water)” as “dsRNA (0.167 mg/ml, in water)”.

Line 287: Correct the following typos “reassociated capsids were stable more than year in the PBS” as “reassociated capsids were stable more than a year in PBS”.

Line 295: It should be used a comma after “In case of HBc/D” to read it as “In case of HBc/D, packaging was achieved…”.

Line 394: Check the format of reference 3. May be the publication date “Published 2015 May 1” could be deleted.

Reference 39: There is no citation in the text for this reference.

Author Response

Line 101: “Region encoding aa 1-195 of HBc from genotype G 99 (GeneBank accession number JQ707677) was amplified using oligonucleotide 5'-GTGGAATTGGCCATGGATAGA-3' as forward primer.” It is not reported the reverse primer. Specify if it is the same primer used to amplify HBc for genotypes A, E and F or write the oligonucleotide sequence.

The specification of reverse primer was added.

and as reverse primer – the primer used to amplify HBc of genotypes A, E and F

Line 113: A full stop is missed at the end of the sentence.

Missed full stop was added.

Line 118: The comma at the end of the sentence should be corrected with a full stop.

The comma was changed to full stop.

Line 126: Correct the following sentence “The expression cultures were incubated fat 37° for 14–16 h…” as “The expression cultures were incubated at 37°C for 14–16 h…”.

The sentence was corrected.

Line 136: Concentration of DNAse I is expressed as [γg/ml]. Considering that 1γg is equal to 1μg could be better to change it to [μg/ml] that is more widely used.

[γg/ml] was changed to [μg/ml].

Line 137- 144: The purification of extracted VLPs by several washing steps (incubation and centrifugation) is difficult to follow. Report the procedure using an easier for readers.

We fully agree with this comment. The purification procedure was rewritten and we believe that procedure of HBc purification became clearer now. The new text:

The purification of HBc VLPs was generally performed through the combination of gel filtration on a Sepharose 4FF column and ion exchange chromatography on a Fractogel DEAE (M) column. 8 g of frozen (-20 °C) biomass was thawed on ice and resuspended in four volumes of lysis buffer (50 mM TrisHCl pH 8.0, 5 mM EDTA, 0.5 mM PMSF, 150 mM NaCl, 0.1% Triton X100 and 5 mM DTT) and disintegrated using a French Press (FrenchPress, GlenMills Inc., USA) at 20,000 psi for three cycles. The total yield of the target protein was verified using SDS-PAGE. For a better outcome of the VLP, urea was added up to 0.5 M to the cell lysate, followed by incubation with DNaseI (125 µg/mL) and MgCl2 to 10 mM at +4 °C for 30 min. Clarification was carried out at 10,000 rpm (13,000 xg), +4 °C for 30 min. To the supernatant ammonium sulphate was added to 10 % of saturation followed by incubation on a rotator at +4 °C for 1 h. After clarification at 10,000 rpm (13,000 xg), +4 °C for 30 min an ammonium sulphate was added to 35% of saturation to supernatant followed by incubation on a rotator at +4 °C for 1 h. After centrifugation at 10,000 rpm (13,000 xg), +4 °C for 30 min, the pellet was dissolved in PBS with 0.5 M urea, 0.5 mM PMSF, 0.1% Triton X100 and 5 mM DTT. Before loading on a Sepharose 4FF 320 mL column, the solution was clarified at 10,000 rpm (13,000 xg), +4 °C for 30 min. The fractions containing VLPs were loaded on an ion exchange Fractogel DEAE (M) 60 mL column equilibrated with PBS/5 mM DTT. The VLPs were eluted from the column in a gradient of 1M KCl. The fractions were analyzed using SDS-PAGE and native agarose gel electrophoresis (NAGE).

Line 149: The author reports that VLPs in PBS were used for electron microscopic analysis. Specify how long after elution of the VLPs the EM analysis was performed or if the VLPs were fixed with glutaraldehyde to avoid alteration of the structure.

We are thankful for this comment.

We did not fix VLPs with glutaraldehyde. We made additions to the description of EM analysis adding the following sentence at the beginning of the section 2.4:

VLPs containing fractions from the column or VLP preparations in PBS were subjected to electron microscopy (EM) analysis immediately or after storage at +4 °C for 1-2 days.

Line 160: A full stop is missed at the end of the sentence.

Full stop was added.

Line 163: Delete the round brackets before the comma.

Brackets were removed.

Line 177: Correct the following typo “reasociation” as “reassociation”.

Correction was made.

Line 176: Paragraph 2.7 "Reassociation of HBc VLPs" is only one and a half lines long. Perhaps you should add it to paragraph 2.6 and make a single paragraph entitled "Dissociation/reassociation of HBc VLPs".

Paragraphs 2.6 and 2.7 were combined making paragraph 2.6. "Dissociation/reassociation of HBc VLPs". Numeration of paragraphs 2.8 and 2.9 were changed to 2.7 and 2.8 accordingly

Line 196: Correct the following typo “dsRNA (0.167 mg/ml, of water)” as “dsRNA (0.167 mg/ml, in water)”.

dsRNA (0.167 mg/ml, of water) was changed to dsRNA (0.167 mg/mL, in water).

Line 287: Correct the following typos “reassociated capsids were stable more than year in the PBS” as “reassociated capsids were stable more than a year in PBS”.

Correction was done.

Line 295: It should be used a comma after “In case of HBc/D” to read it as “In case of HBc/D, packaging was achieved”.

Comma was inserted.

Line 394: Check the format of reference 3. May be the publication date “Published 2015 May 1” could be deleted.

Format of the reference 3 was changed: “Published 2015 May 1” was deleted.

Reference 39: There is no citation in the text for this reference.

Reference 39 was inserted after the reference 38

Reviewer 2 Report

In this paper, Petrovskis et al. analyzed virus-like particles (VLPs) that consist of HBV core proteins of genotypes A-G expressed in E. coli. The results showed that the purified VLP amount was the best in genotypes D and G, and that ss/ds RNA was packed successfully in the particles. The results are important for the possible application of VLPs in gene therapies or vaccines. I have some specific comments to be addressed as below.

  1. Page 8, ‘Electron microscopy revealed the domination of T4 particles in all cases of expressed and purified HBc VLPs.’ – How did the authors distinguish T4 from T3? Although it is described that the T4 particles had a diameter of about 38-40 nm, Fig. 3 shows the diameter is below 30 nm.
  2. Figure 6 and 7, the level of RNA contained in VLPs are evaluated with EtBr staining. However, it is unclear whether the targeted RNA is contained successfully. Please quantify the targeted RNA in VLPs with qPCR and compare genotype D and G.
  3. Figure 3, the labels in panel C are hard to understand. Please remake them.
  4. Table 2, what does ‘Dimer content’ indicate? It is unclear how to calculate it. Please explain clearly.
  5. Some sentences lack periods.
  6. Page 3, parentheses are not used correctly.

Author Response

We are grateful for a thorough review of our manuscript. We have tried to answer all the questions of Reviewers, both in the letter enclosed, and in the text of the manuscript. We have also tried to answer and explain our position when the suggested changes could not be made for technical reasons. The reviews helped us to improve the manuscript substantially and we hope the reviewers will find the changes satisfying and manuscript acceptable for publication in MDPI Microorganisms. The manuscript went through the professional English editing.

Answers to Reviewer comments

  1. Page 8, ‘Electron microscopy revealed the domination of T4 particles in all cases of expressed and purified HBc VLPs.’ – How did the authors distinguish T4 from T3? Although it is described that the T4 particles had a diameter of about 38-40 nm, Fig. 3 shows the diameter is below 30 nm.

We are thankful for this comment. We agree that Figure 3 causes confusion - from EM pictures one can indeed conclude that HBc particle size is around 30 nm. We apologize for this misunderstanding and want to explain that EM pictures in Results are given in this case only for the demonstration of particle morphology, not for exact size determination. Therefore we made several changes in the text:

  1. Materials and Methods paragraph 2.4 starts now with this addition:

“VLPs containing fractions from the column or VLP preparations in PBS were subjected to electron microscopy (EM) analysis immediately or after storage at +4 °C for 1-2 days. VLPs samples”,

  1. to the end of paragraph 2.4 the folowing has been added:ImageJ program was used for measurements of precise HBc particle size”.
  2.  
  3. to the end of the Results, paragraph 3.1 has been added:

“Electron microscopy revealed the domination of T4 particles in all cases of expressed and purified HBc VLPs. HBc particle size using ImageJ program was on average 34.4 nm for ten T4 particles and 27.75 nm for ten T3 particles. The hydrodynamic diameter of particles detected using DLS s was found to be around 38 nm”.

  1. Figure 6 and 7, the level of RNA contained in VLPs are evaluated with EtBr staining. However, it is unclear whether the targeted RNA is contained successfully. Please quantify the targeted RNA in VLPs with qPCR and compare genotype D and G.

We have not made RNA quantification in this study by technical reasons.

Our explanation: As seen in Figures 6 and 7, both ss and ds RNA are incorporated in HBc VLPs at the level giving good EtBr staining signal. However, since the signal of incorporated RNA is comparable to the signal of purified native HBc VLPs containing RNA from the expression host (compare tracks 6 and 7of Figure 6 (b)), we conclude that target RNA is incorporated successfully and effectively. Also, as shown in Figure 6, the level of dsRNA incorporation for D and G HBc genotypes is similar; - the same was observed for ssRNA incorporation using RNA from phages PP7 (shown only for D genotype, Figure 7). Note: we have found the mistake in the description of the tracks 6 and 8 in Figure 6, therefore the following changes have been made in the legend of Figure 6:

6-9 - NAGE of HBc/G VLPs 6 – native HBc/G VLPs, 7 - packaged with dsRNA HBc/G VLPs, stained with EtBr, and 8 – native HBc/G VLPs, 9 – packaged HBc/G VLPs stained with Coomassie Brilliant Blue R-250. E, EM of HBc/G VLPs packaged with dsRNA. F, DLS of HBc/G VLPs packaged with dsRNA.

  1. Figure 3, the labels in panel C are hard to understand. Please remake them.

Labels in panel C of Figure 3 have been changed.

  1. Table 2, what does ‘Dimer content’ indicate? It is unclear how to calculate it. Please explain clearly.

We absolutely agree with the reviewer that dimer outcome was not clearly described. We made the following changes:

  1. we supplemented the Materials and Methods, paragraph 2.6 adding the following:

“Outcome of dimers was calculated as a proportion of HBc material in dimer fractions after analysis of dissociation products on Superose6 and amount of HBc VLPs used for dissociation”.

  1. Results, section 3.2 was rewritten in this way:

HBc/D, HBc/G, and HBc/F VLPs were selected for the dissociation–reassociation of HBc VLPs. Here, VLPs were dissociated using dissociation buffer containing GuHCl and LiCl. After incubation in the GuHCl containing buffer part of VLPs went to debris. After removal of debris the supernatant was loaded on Superdex200 column to separate dimers from the non-dissociated capsids and other impurities. Results of representative separation are shown in Figure 4. The results showed that supernatant contained mostly dimers (Figure 4). The most effective dissociation was observed for HBc/G (80% of of treated soluble VLPs in dimers), followed by HBc/D (70%) and HBc/F (40%).”

  • to the legend of Figure 4 was added the following:

“…and removal of insoluble material.”

  1. Some sentences lack periods.

Full stops were inserted.

  1. Page 3, parentheses are not used correctly.

Use of parentheses has been corrected.

Round 2

Reviewer 2 Report

In this revised paper, Petrovskis et al. addressed the issues and modified the manuscript substantially. However, I still have a concern as below.

  1. Figures 6 and 7. The authors did not confirm that the targeted RNA was contained in VLPs specifically. They claimed that RNA quantification with qPCR is not needed, but I think another method is required. If the RNA that is extracted from the VLPs is electrophoresed in an agarose gel, a band corresponding to the targeted ss/dsRNA will be made. Also, in the reassociation step, a negative control without ss/dsRNA is required. I think these are not technically difficult.

Author Response

We are grateful to You for the thorough and careful examination of our manuscript.

Our answer:

  1. Concerning the packaging of dsRNA.

In full agreement with the Reviewer, that examination of the incorporated RNA is necessary, we would like to explain that appropriate tests in fact were done by us in the course of this study. Concerning the incoroprated dsRNA, we have modified the manuscript including additional data:

  • We have added Figure 8 with the data about dsRNA extraction from reassociated HBc/D VLPs containing packaged dsRNA. Also, as requested by the Reviewer, negative control – empty VLPs, without RNA obtained after reassociation of dimers at the absence of RNA, is included (Figure 8, tracks 6,7).

Specifically, the text have been supplemented with the following new content:

  • Paragraph 3.4 has been added to Results:

3.4. Analysis of RNA packaged in HBc VLPs

 For analysis of nucleic acid content of HBc VLPs, native HBc/D particles and HBc/D particles obtained after reassociation of dimers at the presence of dsRNA were treated by Proteinase K to disrupt particles (Materials and Methods, 2.9.). After deproteinization the obtained nucleic acids were analyzed by NAGE. As shown in Figure 8, RNA both native and packaged particles are extracted in comparable amounts (tracks 2 and 3). Similar results were obtained for HBc/G packed with dsRNA (not shown). Particles obtained by reassociation of dimers are free of RNA (track 6, no signal), and these particles are not able to enter the gel, staying on the start (track 7).

  • Figure 8. Extraction of RNA from HBc/D VLPs, NAGE in 2% Agarose. M – GeneRuller 1 kb DNA Ladder (SM1331, Thermo Scientific™), 1 – dsRNA used for packaging, 2 – RNA extracted from packaged HBc/D VLPs, 3 – RNA extracted from native HBc/D VLPs, 4 – native HBc/D VLPs, 5 – HBc/D VLPs packaged with dsRNA, 6, 7 – empty HBc/D VLPs; M and 1-6 – stained with EtBr, 7 – stained with Coomassie Brilliant Blue R250. Note: Different amounts of VLPs are loaded in tracks 4 and 6, not corresponding to the amount of VLPs used for the extraction of RNA (tracks 2 and 3).
  • Materials and Methods have been supplemented with additional section 2.9:

 2.9. Extraction of RNA from HBc VLPs

For extraction of RNA from HBc particles (VLPs) disruption of particles with Proteinase K was used. Briefly, to 160 µg of native or packaged with dsRNA HBc/D VLP protein solution, SDS up to 0.5% and 10 µL of Proteinase K (Thermo Scientific EO0491, Lithuania) were added and mix was incubated at 37 °C overnight. The samples were deproteinized by chloroform and RNA was dissolved in 20 µL of dH2O. Ten µL of the solution was used for analysis by NAGE.

  • Concerning the packaging of ssRNA.We fully agree with the Reviewer that this is of big importance to see also the content of VLPs packaged with ssRNA. To test the functionality of incorporated ssRNA we extracted RNA from HBc particles packaged with phage RNA (“ssRNA”) and used the extracted RNA to transfect E.coli cells. We describe in a separate paper (see Discussion) that these extracted ssRNA molecules were able to establish specific RNA-phage infection (manuscript prepared for submission, Lieknina, I. et al. [50]).
  1.  

Round 3

Reviewer 2 Report

The authors added a new data to address the raised issue. The manuscript has been improved.